


**Conversion relationships between Modified Mercalli Intensity and Peak Ground**
**Acceleration for historical shallow crustal earthquakes in Mexico**
**Quetzalcoatl Rodríguez-Pérez[1,2], F. Ramón Zúñiga[2]**
[1] Consejo Nacional de Humanidades, Ciencia y Tecnología, Mexico City, Mexico.
[2] Instituto de Geociencias, Universidad Nacional Autónoma de México, Juriquilla, Querétaro, Mexico.
**Correspondence:** Quetzalcoatl Rodríguez-Pérez (quetza@geociencias.unam.mx)
**Abstract.** New empirical relationships between modified Mercalli intensity (MMI) and synthetic peak
ground acceleration (PGA) are developed for shallow crustal earthquakes in Mexico. Ground motion
data from 18 moderate-to-large earthquakes ($4.5 < M_W < 7.5$) and the corresponding 531 MMI
information reports were employed. Synthetic PGA data were generated using the finite-fault stochastic
method considering different rupture scenarios in order to extend the limitations of the dataset. Linear
and bilinear regression techniques were used considering a binning averaging procedure and the whole
dataset. On one hand, a set of MMI predictive equations independent of moment magnitude ($M_W$) and
hypocentral distance ($R$) were derived. Despite weak dependencies of the residuals on $M_W$ and $R$ terms,
on the other hand, we also developed refined predictive relationships that include these parameters as
independent variables. The refined PGA to MMI conversion equations show slightly less variability
than simple linear equations in predicting intensity values. The proposed predictive equations are
consistent with similar relationships in other regions of the world. The discrepancies among the
different relationships may reflect the differences in input data, particularly related to the macroseismic
intensity assignments, which are inherently subjective, and the tectonic regime. The proposed
relationships can be used for improved hazard assessments in Mexico.



# 1 Introduction

Seismic intensity is a measurement that describes the degree of shaking or damage at a particular site. Usually, seismic intensity is represented spatially by intensity maps. These maps provide shaking patterns reflecting earthquake damage and its effects on human dwelings from a specific earthquake. The intensity scale was originally designed to describe and quantify seismic damage in the absence of instrumental recordings. Arguably, one of the first attempts to describe effects of earthquake shaking on a map which led to the development of the intensity scales, was carried out by Mallet (1862). Several intensity scales have been proposed throughout time, for example, the European Macroseismic Scale, the Japan Meteorological Agency seismic intensity scale (JMA), the China Seismic Intensity Scale, the Mercalli-Cancani-Sieberg scale, and the Modified Mercalli Intensity scale (MMI). MMI is currently the most widely used scale worldwide. MMI quantifies the effects of an earthquake using 12 degrees, with 1 indicating not felt and 12 as complete destruction (Wood and Neumann, 1931). The intensity values vary with the distance to the earthquake source, with the highest intensities usually around the epicentral region and based on subjective information collected from persons who have felt the earthquake at a specific location. Seismic intensity observations in Mexico have been reported in the literature for particular earthquakes (mainly historical events) (e.g., DuBois and Smith, 1980; Suter et al., 1996; Suter, 2015a, 2018, 2019, 2020; Suter and Morelos-Rodríguez, 2023) or in comprehensive earthquake catalogs (Figueroa, 1963, 1986; Suárez, 2021).

Several efforts have been made to develop relationships between seismic intensity and ground motion parameters such as the peak ground acceleration (PGA), peak ground velocity (PGV), peak ground displacement (PGD), seismic magnitude ($M$), and hypocentral distance ($R$) among others (Trifunac and Brady, 1975; Murphy and O'Brien, 1977). In the absence of ground motion observations, various


approaches to model ground motion parameters for different rupture scenarios for historical
earthquakes have been explored, including dynamic rupture modeling, kinematic modeling, modeling
with ground-motion prediction equations, and stochastic modeling. Various prediction relations from
ground motion parameters have been proposed for different regions; for instance, for California (Wald
et al., 1999; Atkinson and Sonley, 2000; Atkinson and Kaka, 2007), Taiwan (Wu et al., 2003), Greece
(Tselentis and Danciu, 2008), and Turkey (Bilal and Askan, 2014), among other zones. The correlation
relationships can be developed in different forms, such as using regional, global, or synthetic data. In
this article, we determined conversion relationships between seismic intensities and PGA for 18
historical shallow crustal earthquakes which took place in Mexico with moment magnitudes in the
range of 4.5 – 7.5. The seismic events occurred between 1568 and 1989, and seismic intensities were
reported in the literature. Since the last $M \geq 6$ event originated in north and central Mexico (we exclude
the strike slip regime of northern Baja California Peninsula in this context) took place in 1920, we
cannot rely on observed records alone. Therefore, we rely on PGA values generated by synthetic
accelerograms derived from the finite-fault stochastic method (Motazedian and Atkinson, 2005). The
calculated relationships will be relevant for various purposes ranging from assessment of historical
seismicity to several applications in seismic hazard and risk studies from shallow crustal earthquakes in
Mexico.
**2 Tectonic setting**
The epicenters of the studied earthquakes occurred in three tectonic provinces: 1) Basin and Range
province (BRP, two events), 2) Sierra Madre Oriental fold-thrust belt (SMOFTB, two events), and 3)
the Trans-Mexican Volcanic Belt (TMVB, 14 events) (Fig. 1). The BRP is characterized by the
alternance of north-south trending mountains and flat valley floors comprising the province. The 5
March 1887 Bavispe earthquake ($M_W$ = 7.5) occurred in a north-south striking west-dipping normal





fault system distributed on the western limit of the Sierra Madre Occidental. This event is considered
the largest crustal fault earthquake in Mexico (Suter, 2015b). The rupture has been modeled with a dip
angle of 70° and a maximum displacement of 5.2 m and comprises three fault segments (the Pitáycachi,
Teras, and Otates segments, respectively) (Suter, 2015b). The second event in the BRP at the Bolaños
graben was the 6 November 1774 event  ($M_\mathrm{W}$ = 5.7) (Suter, 2020). The SMOFTB is the most prominent
tectonic characteristic in eastern Mexico and constitutes the continuation to the south of the North
American Rocky Mountains. The southern part of the SMOFTB and the northern part of the TMVB are
being deformed by mostly north-south-striking extension structures. Two studied earthquakes occurred
in this geological structure at a latitude of about 21° in east-central Mexico in a segment 130 km length
along the Moctezuma River (Suter, 1987). The TMVB is an active volcanic arc associated with the
subduction of oceanic plates along the Pacific margin of Mexico. The TMVB spans over 1200 km,
from the Pacific Ocean to the Gulf of Mexico, and is 100 km wide (Ferrari et al., 2012). The western
part of the TMVB exhibits three major normal fault systems oriented: NW-SE, NNE-SSW, and WNW-
ESE. The central part of the TMVB is characterized by normal faults with a left-lateral strike-slip
component orientated east-west. The eastern segment of the central TMVB is dominated by a left-
lateral transtensive deformation, whose extension is oriented NW-SE (Ego and Ansan, 2002).
**3 Data and methods**
We studied ground motion characteristics of 18 historical shallow crustal earthquakes in north and
central Mexico with magnitudes in the range of 4.5 – 7.5 (Table 1 and Fig. 1). For this purpose, we
used reported seismic intensities (531 MMI observations) in conjunction with synthetic PGA values to
derive conversion relationships. Reported MMI values varied from 2 to 11 recorded at hypocentral
distances in the range of 8.14 – 1800 km (Fig. 2).

## 3.1 Stochastic finite-fault method

The stochastic strong-ground motion method was proposed by Boore (1983) as a fast and reliable

technique for simulating high-frequency characteristics generated by earthquakes. The method has been

subsequently improved by incorporating different sources (point or finite-fault models), paths, and site

effects (H/V curves or theoretical functions) to calculate synthetic signals (displacement, velocity, or

acceleration signals) to be contrasted then with the observed records. We used the stochastic finite-fault

ground motion approach developed by Motazedian and Atkinson (2005) as modified by Atkinson and

Assatourians (2015) to model ground motions of the selected events. Following that approach, the fault

plane is subdivided into a grid of subsources (or subfaults), assigning a stochastic point source to each

subfault. Each subfault is activated once with an appropriate delay time. A point source Brune $\omega^2$-

source spectrum (Brune, 1970) is generated for each subfault. The point source spectrum is derived by

multiplying the source, path, and site spectra in the frequency domain (Boore, 1983; 2003). The

acceleration spectrum of a point source is described by the equation

$$Y(M_0, R, f) = A(M_0, f) P(R, f) S(f) \quad , \tag{1}$$

where $A(M_0, f)$, $S(f)$, and $P(R, f)$ represent the source, site, and path spectra, respectively. $M_0$ is the

seismic moment, $R$ is the hypocentral distance, and $f$ is the frequency. The acceleration spectra after

considering the Brune model is (Motazedian and Atkinson, 2005)

$$A(M_0, f) \frac{R_\theta \ VF M_0 H_{ij} (2\pi f)^2}{4\pi \rho \beta^3} G(r) \frac{1}{\left[1 + (f/f_c)^2\right]} e^{-\pi f R_{ij}/Q(f)\beta} Z(f) e^{-\pi \kappa f} \quad , \tag{2}$$

where $R_{\theta\varphi}$ is the radiation pattern (0.55), $F$ is the free surface amplification (2.0), $V$ is the partition of





energy into two horizontal components (0.7), $\rho$ is the rock density, $\beta$ is the $S$-wave velocity, $f_c$ is the
corner frequency, $M_0$ is the seismic moment, $H_{ij}$ is a frequency-dependent scaling factor for high-
frequencies, $G(R)$ is the geometrical spreading function, $Q(f)$ is the seismic attenuation factor, $R_{ij}$ is the
distance from the observation point to the $ij^{th}$ subfault, and $Z(f)$ is a function that describes site
amplification. The corner frequency is defined as $f_c = N(t)^{-1/3}\ 4.9\ x\ 10^6\ \beta\ (\Delta\sigma/M_{0ave})^{1/3}$, where $\Delta\sigma$ is the
stress drop, $M_{0ave}$ is the average seismic moment of the fault ($M_{0ave} = M_0/N$, here $N$ is the number of
subfaults), and $N(t)$ denotes the cumulative number of all ruptured subfaults at time $t$. The acceleration
spectra ($A(M_0,f)$) is combined with random phases and transferred into the time domain for each point
source on the finite-fault plane. The contribution from each subfault is summed up from the entire fault
in the time domain with the following equation ($a(t)$)

$$a(t) = \sum_{i=1}^{nl} \sum_{j=1}^{nw} a_{ij}(t - \Delta t_{ij} - T_{ij}) \quad , \tag{3}$$

where $nl$ is the subfault number along the fault length, $nw$ is the same along the fault width, $\Delta t_{ij}$ is the
relative time delay that the radiated waves from the $ij^{th}$ subfault arrive at the observation point, and $T_{ij}$ is
the fraction of rise time. The duration of motion comes from the source duration plus the path duration.
The ground motion simulations were performed with the code EXSIM12 (Assatourians and Atkinson,

18 2012).

**3.2 Finite-fault stochastic ground motion simulation input parameters**
For the analyzed events, we used moment magnitude. In some cases, it was taken from previous studies
(Suter, 2015b; Suárez et al., 2019), but in other cases, it was derived from conversion relationships
between $m_b$ and $M_W$ estimated by Scordilis (2006). Seismic velocities and densities at hypocenter





locations were taken from the 3D elastic model of Mexico derived by Spica et al. (2016). For the case
of earthquakes along the TMVB, we used frequency-dependent attenuation models determined by
Pérez-Moreno et al. (2021). On the other hand, the seismic attenuation model for the Sonora region
(Castro et al., 2008) was used to simulate the March 5, 1887, Bavispe earthquake. Previous studies
showed that the spectral decay parameter is in the range of 0.03 – 0.04 sec for crustal earthquakes in
Mexico (e.g., Fernandez et al., 2009; Lermo et al., 2016). We calculated fault dimension with the
scaling relations for normal-faulting earthquakes following Thingbaijam et al. (2017). Strike and dip
angles for the studied events are reported in Table 2. Synthetic slip distributions were generated with
the slip spatial random field model proposed by Mai and Beroza (2002) using a von Karman
autocorrelation function. We used different stress drop values to perform stochastic ground motion
simulations to calculate PGA values (1, 5, 10, and 20 MPa). Site effects were both quantified by
generic rock and soil site amplification functions of Boore and Joyner (1997). We used the geometrical
spreading and distance-dependent path duration model for eastern North America  based on shallow
crustal earthquakes proposed by Atkinson and Boore (1995).
**3.3 Regression analysis**
After obtaining synthetic peak ground acceleration/reported intensity pairs, we fitted the data
considering two distinct forms: 1) using the complete dataset and 2) averaging the data by different
MMI levels as commonly presented by several authors (e.g., Atkinson and Kaka, 2007; Worden et al.,
2012). The latter approach is used to obtain regression results that are stable and well-constrained for
each intensity level. The relationship between PGA and intensity pairs is commonly represented by
linear models (Wald et al., 1999; Atkinson and Sonley, 2000). As previously reported, inspecting the
complete dataset exhibited a two-branched trend (Figs. 3 and 4). In this case, we fitted the data using
piecewise linear fitting Python code PWLF (Jekel and Venter, 2019). We adopted an approach similar





to Atkinson and Kaka (2007) to determine the averaged linear model. We started by finding the mean
$\log_{10}$PGA for each MMI level (2 to 10), including in each average the data for MMI 2.0 – 3.75, 3.76 –
4.49, 4.50 – 5.49, 5.50 – 6.49, 6.50 – 7.49, 7.50 – 8.49, 8.50 – 9.49, 9.50 – 10.49, and 10.50 – 11.49.
The result followed a linear trend (Figs. 3 and 4). We regressed MMI against the $\log_{10}$ PGA to obtain
the bilinear predictive equations of the form (model 1, dark blue lines in Figs. 3 and 4)
$MMI = c_1 + c_2 \log_{10} PGA$    for $\log_{10}$ PGA $\le t_1$,      (4)
$MMI = c_3 + c_4 \log_{10} PGA$    for $\log_{10}$ PGA $> t_1$
where $c_1$ to $c_4$ are the fitted parameters, and $t_1$ is intersection of the two lines. On the other hand, the
single linear model is here referred to as model 2 (red lines in Figs. 3 and 4). Table 3 summarizes the
parameters for each of the regression models. Following Atkinson and Kaka (2007), we applied a
correction term to the regressions previously described. This term accounts for magnitude and distance
effects and is defined as
$\Delta_{MMI} = c_5 + c_6 M_W + c_7 \log_{10} R$    for $\log_{10}$ PGA $\le t_1$,      (5)
$\Delta_{MMI} = c_8 + c_9 M_W + c_{10} \log_{10} R$    for $\log_{10}$ PGA $> t_1$
where $M_W$ is the moment magnitude and $R$ is the hypocentral distance. The coefficients for these
regressions are given in Table 4. The improved prediction equations of MMI for PGA are the sum of
models 1 to 2 and Eq. (5). In the case of a simple linear model, only one part of Eq. (5) is used.
**4 Results**





The regression coefficients and standard errors based on Eqs. (4) and (5) are listed in Tables 3 and 4.
Residuals as a function of magnitude and distance are shown in Figs. 5 and 6 for the bilinear and linear
models, respectively. We evaluated the variations of residuals versus $M_W$ and hypocentral distance,
observing that there is no obvious dependency in the residuals on these parameters. Our results also
showed that the residuals are smaller for the linear model by a factor of 8 concerning residuals for the
bilinear model. The standard errors of regression equations support this observation; in this case, errors
are smaller by factors between 2 and 3 (Table 3). After applying the correction term to the residuals, we
observed a slight reduction in the standard errors (Table 4). By comparing results for rock and soil site
conditions, our results exhibit similar standard error values (Tables 3 and 4). Our results showed that
MMI-PGA relationships derived for stress drop values of 10 and 20 MPa have the lowest standard
errors. Figures. 7 to 10 show a comparison of the resulting MMI-PGA models with relations obtained
in previous studies for regions with similar tectonic features based on shallow crustal earthquakes.
These are discussed in the following section.
**5 Discussion and conclusions**
We contrasted our results with reported MMI-PGA relations for California (5.8 < M < 7.3, Wald et al.,
1999; 4.9 < $M_W$ < 7.4, Atkinson and Sonley, 2000), Central United States (1.8 < M < 7.1, Atkinson and
Kaka, 2007), Greece (4 < M < 6.9, Tselentis and Danciu, 2008), Iran (5.1 < $M_W$ < 7.3, Ahmadzadeh et
al., 2020), and Türkiye ( 5.7 < $M_W$ < 7.4, Bilal and Askan, 2014) (Figs. 7 to 10). The bilinear models
differ significantly at lower PGA values (< 10 cm/s$^2$). On the other hand, linear models are consistent
with most of the relationships reported in previous studies. The observed discrepancies among the
MMI-PGA relationships may reflect the differences in input data, especially the macroseismic intensity
assignments and the regional tectonic environment.



The standard errors of our linear models reinforce the observation that the binning procedure proposed
by Atkinson and Kaka (2007) provides stable and reliable results. Although the whole data may be
considered without averaging, it gives more significant standard errors. In particular, a stress drop value
of 10 MPa provides a correlation between PGA and MMI in good agreement with global studies.
According to Kanamori and Anderson (1975), a stress drop value of 10 MPa can be considered typical
for intraplate earthquakes. On the contrary, Allmann and Shearer (2009) reported a median stress drop
of 5.95 ± 1 MPa for intraplate events. If we are to follow this assumption, our regression models
determined for a stress drop value of 5 MPa might be the best choice. Our results showed weak
dependencies of the residuals on magnitude or distance terms; we also developed refined predictive
relationships that include these parameters as independent variables.
In Mexico, few strong-motion recordings of shallow crustal earthquakes have led to uncertainties in the
estimation of seismic risk in the region. Estimations of ground motions during these events can be
obtained by converting reported MMI to PGA or PGV using conversion relations derived from other
regions. This, nevertheless, has not been carried out, mainly because no such conversion relationships
were available for Mexico (Córdoba-Montiel et al., 2018). In this sense, we tested our conversion
relationships with other PGA estimations, particularly for the 1920 Jalapa earthquake (event 13 in Table
1). Córdoba-Montiel et al. (2018) determined a PGA for the Jalapa City (MMI = 9, with $\Delta\sigma$ = 5 and 10
MPa) of 100 - 250 cm/s$^2$. Our estimates of PGA are consistent with their results for the same stress
drops (176 < PGA < 300 cm/s$^2$). Our conversion relations are the first MMI-PGA relations proposed for
north and Central Mexico. Overall, our results showed that the stochastic ground motion simulation
method is a valid methodology to determine MMI-PGA relations consistent with those derived with
peak ground motion observations, being a suitable option to add to the study the effects of historical
earthquakes with MMI observations available. The proposed methodology can be improved by



including precise site effect estimations (e.g., horizontal-to-vertical spectral ratio, H/V) instead of
considering generic site effects. The predictive equations herein proposed can  also be used to produce
intensity maps for shallow crustal earthquakes in Mexico.
*Author contributions.* QRP and FRZ designed the framework of the manuscript. QRP performed
ground motions simulations and regression analysis. All the authors discussed the results and wrote the
paper.
*Code availability.* Bilinear regression analysis was performed with the PWLF code
(https://github.com/cjekel/piecewise_linear_fit_py, Jekel and Venter, 2019). We used the software
EXSIM12 to simulate ground motions (https://github.com/GFZ-Centre-for-Early-Warning/exsim,
Assatourians and Atkinson, 2012).
*Competing interests.* The authors declare no competing interests.
*Financial support.* QRP was supported by the Mexican National Council for the Humanities, Science,
and Technology (CONAHCYT) (Catedras program - project 1126).

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

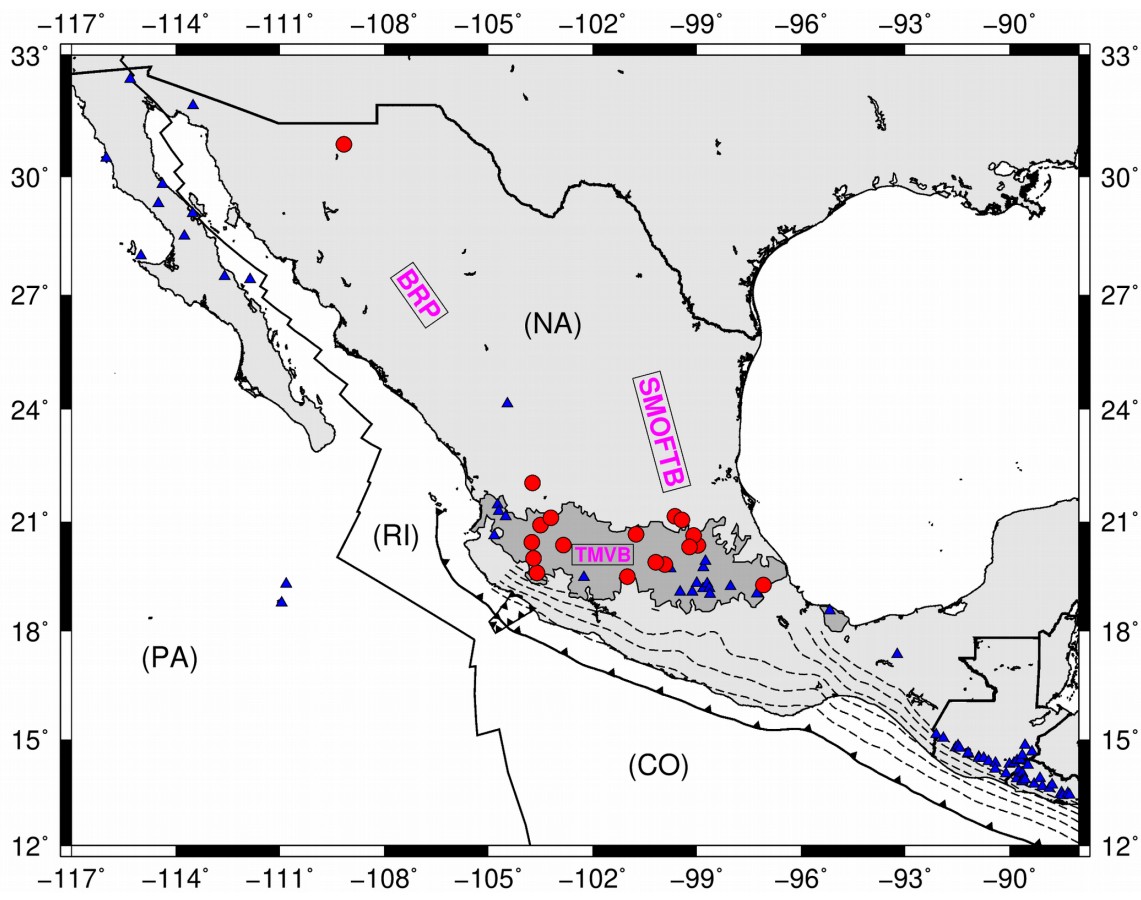

**Figure 1.** Epicenter locations of the studied earthquakes (red dots). Blue triangles are volcanoes. NA, RI, CO, and PA are the North America, Rivera, Cocos and Pacific plates. BRP is the Basin and Range province, SMOFTB is the Sierra Madre Oriental fold-thrust belt, and TMVB is the Trans-Mexican Volcanic Belt.

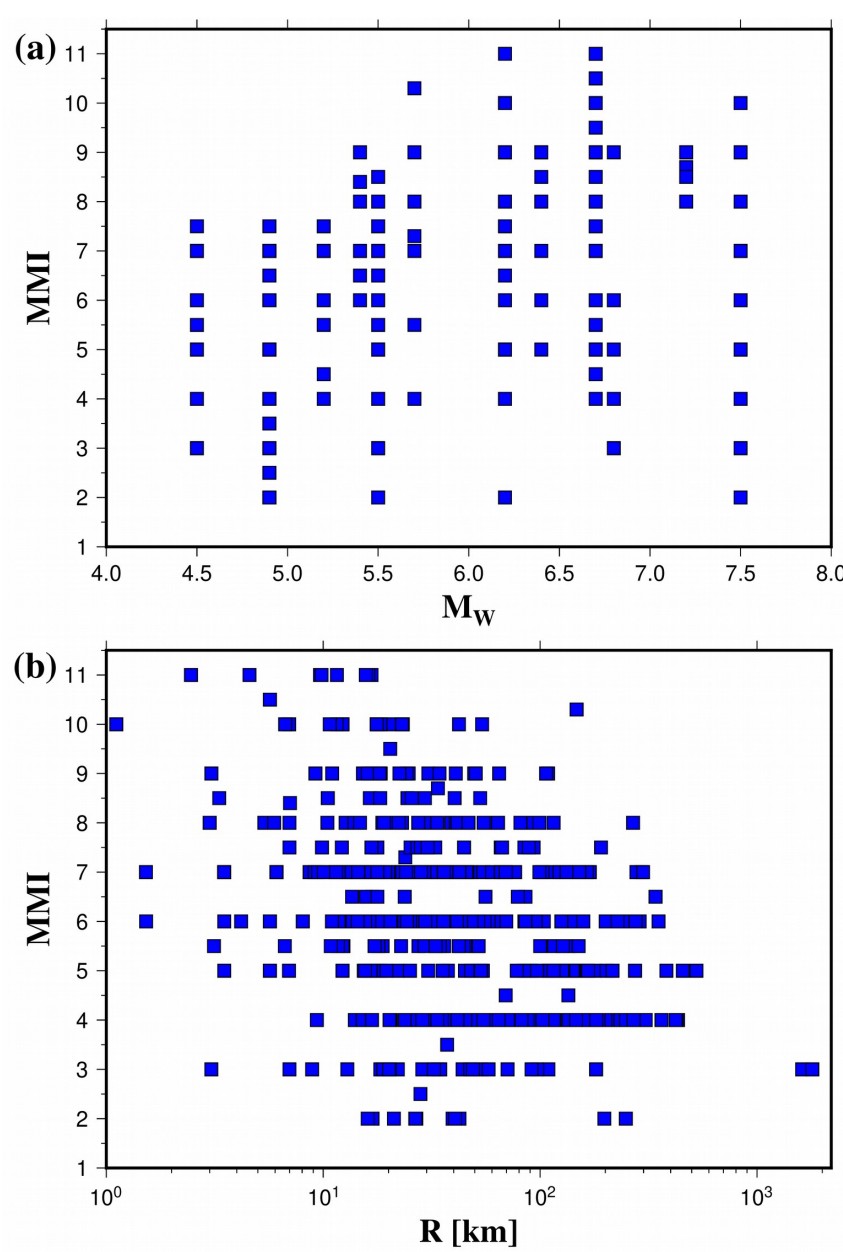

Figure 2. (a) Modified Mercalli intensity (MMI) versus moment magnitude ($M_W$) of historical shallow earthquakes in Mexico. (b) MMI versus hypocentral distance ($R$) of the studied earthquakes.

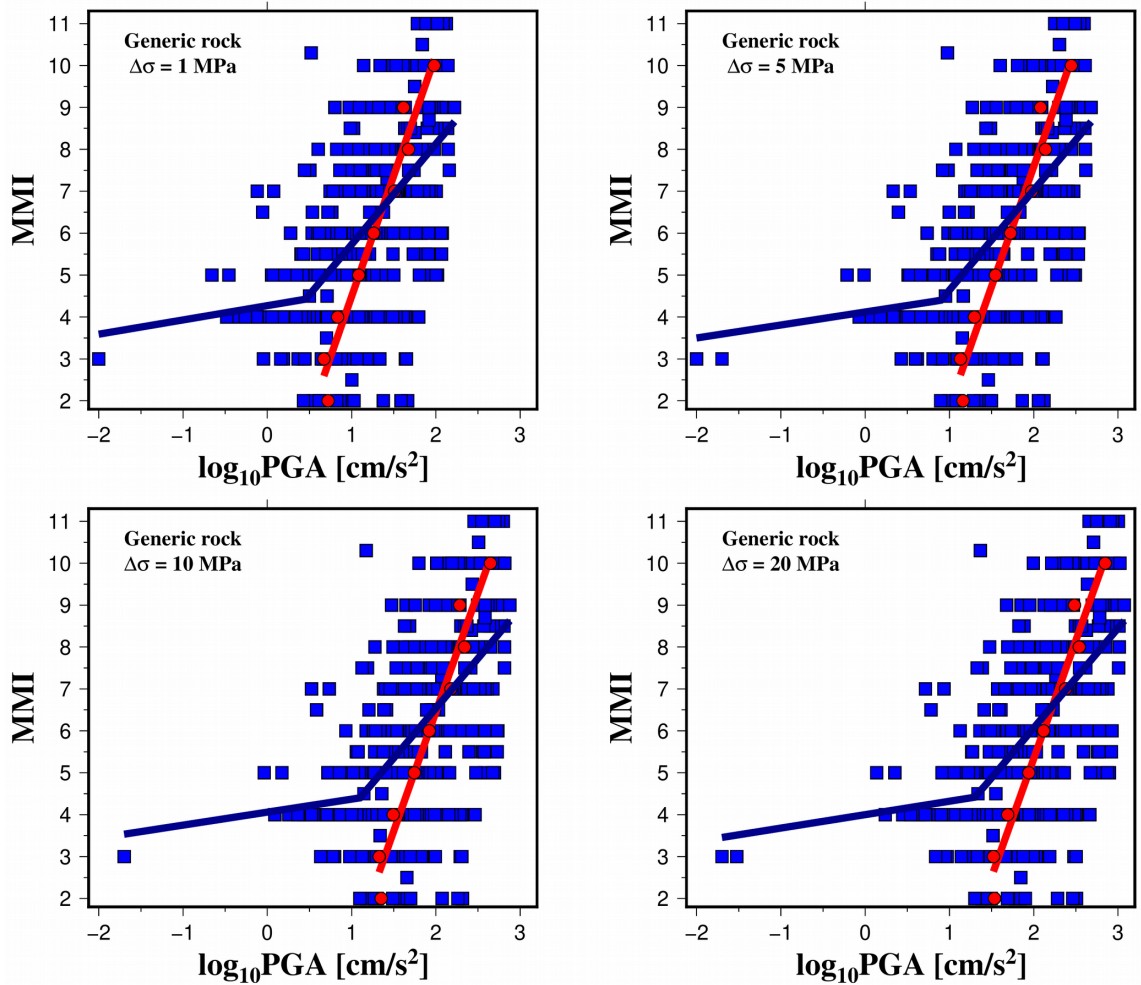

**Figure 3.** Synthetic peak ground acceleration/reported intensity pairs considering generic rock site and

different stress drop conditions ($\Delta\sigma$ = 1, 5, 10, and 20 MPa) (blue squares). Red circles represent mean

$\log_{10}$ PGA values for each MMI level. Predictive relationships derived in this study: bilinear fit (dark

blue lines) and averaged data linear fit (red line).

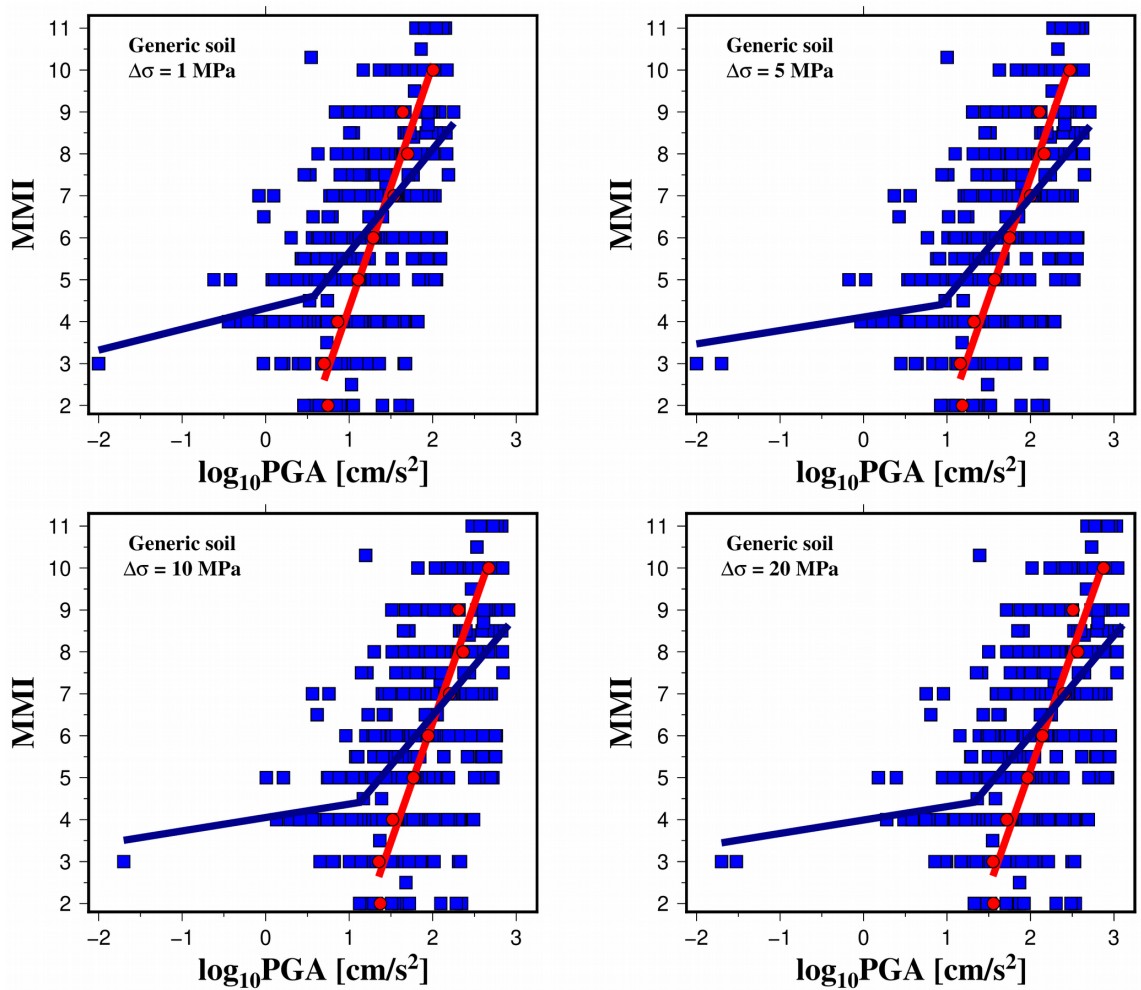

**Figure 4.** Synthetic peak ground acceleration/reported intensity pairs considering generic soil site and different stress drop conditions ($\Delta\sigma$ = 1, 5, 10, and 20 MPa) (blue squares). Red circles represent mean $\log_{10}$ PGA values for each MMI level. Predictive relationships derived in this study: bilinear fit (dark blue lines) and averaged data linear fit (red line).



**Figure 5.** Residuals (observed MMI minus predicted MMI) for MMI predicted from PGA obtained

from the bilinear model as a function of earthquake magnitude **(a)** and the log of hypocentral distance

**(b)**.


**Figure 6.** Residuals (observed MMI minus predicted MMI) for MMI predicted from PGA obtained

from the averaged data linear fit as a function of earthquake magnitude **(a)** and the log of hypocentral

distance **(b)**.



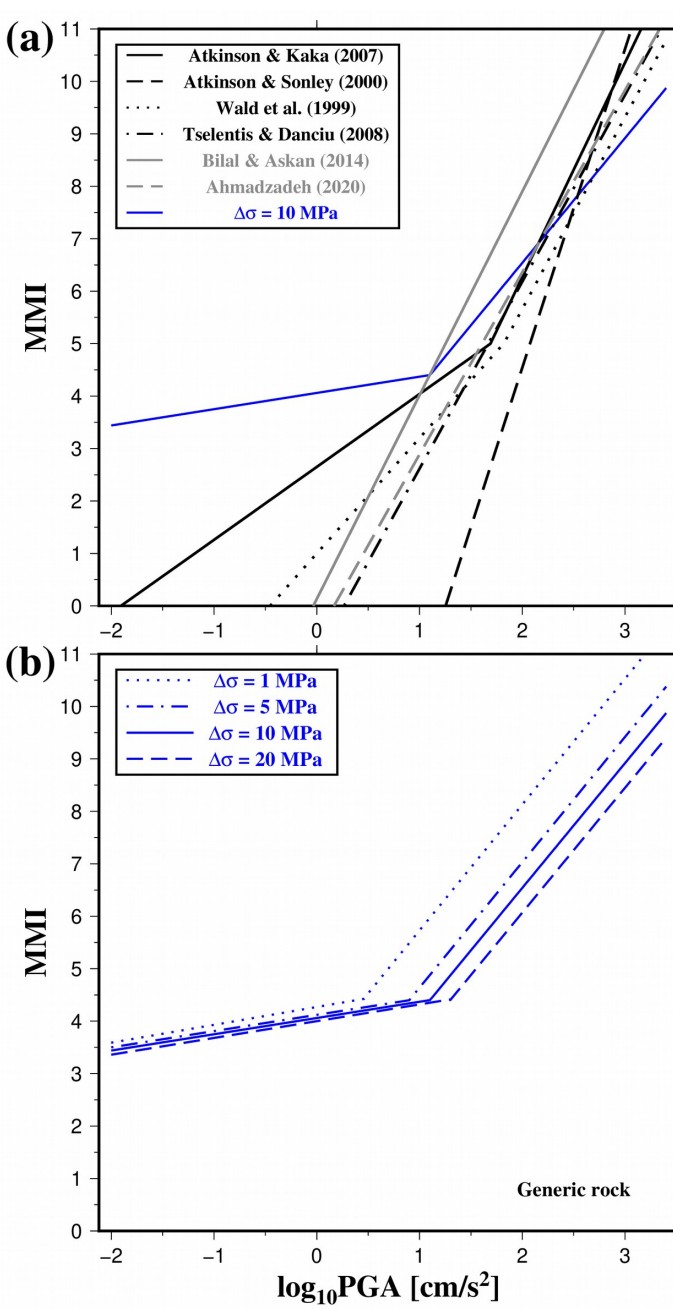

**Figure 7. (a)** Comparison of the obtained MMI-PGA relationships (blue lines) considering generic rock

site conditions with previous studies. **(b)** The lower panel shows results for different stress drop values.


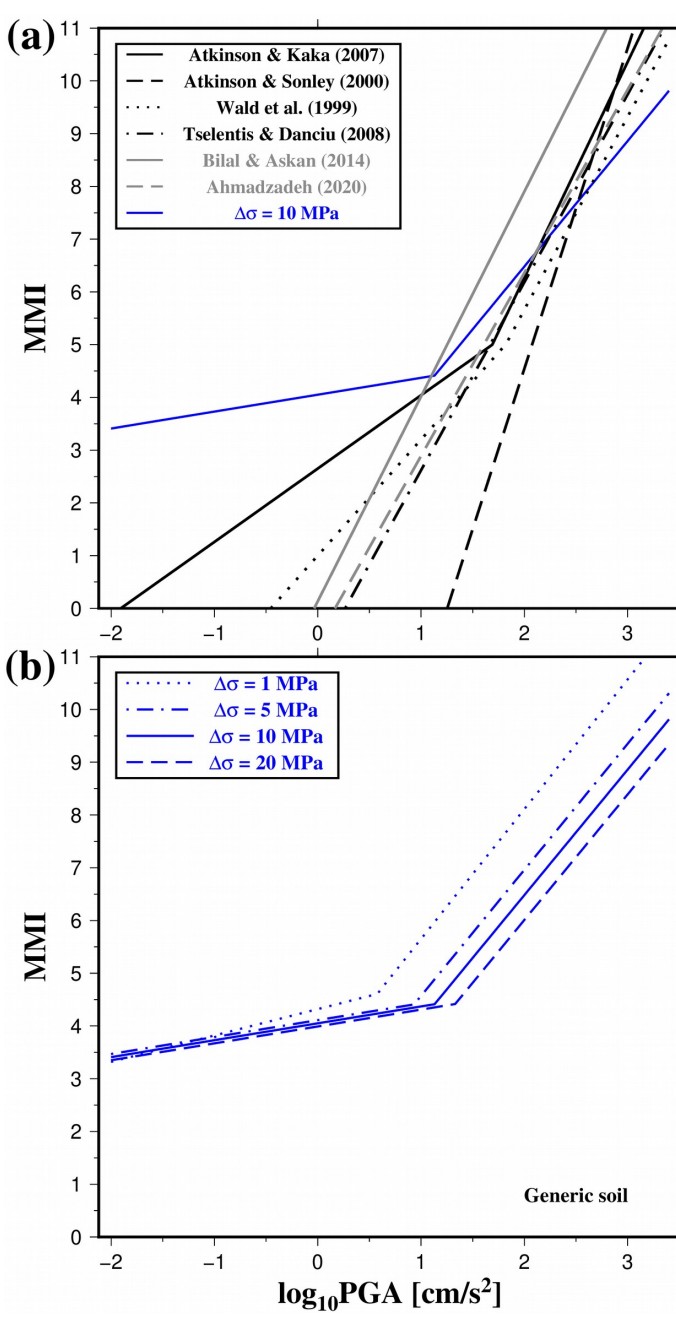

**Figure 8. (a)** Comparison of the obtained MMI-PGA relationships (blue lines) considering generic soil

site conditions with previous studies. **(b)** The lower panel shows results for different stress drop values.



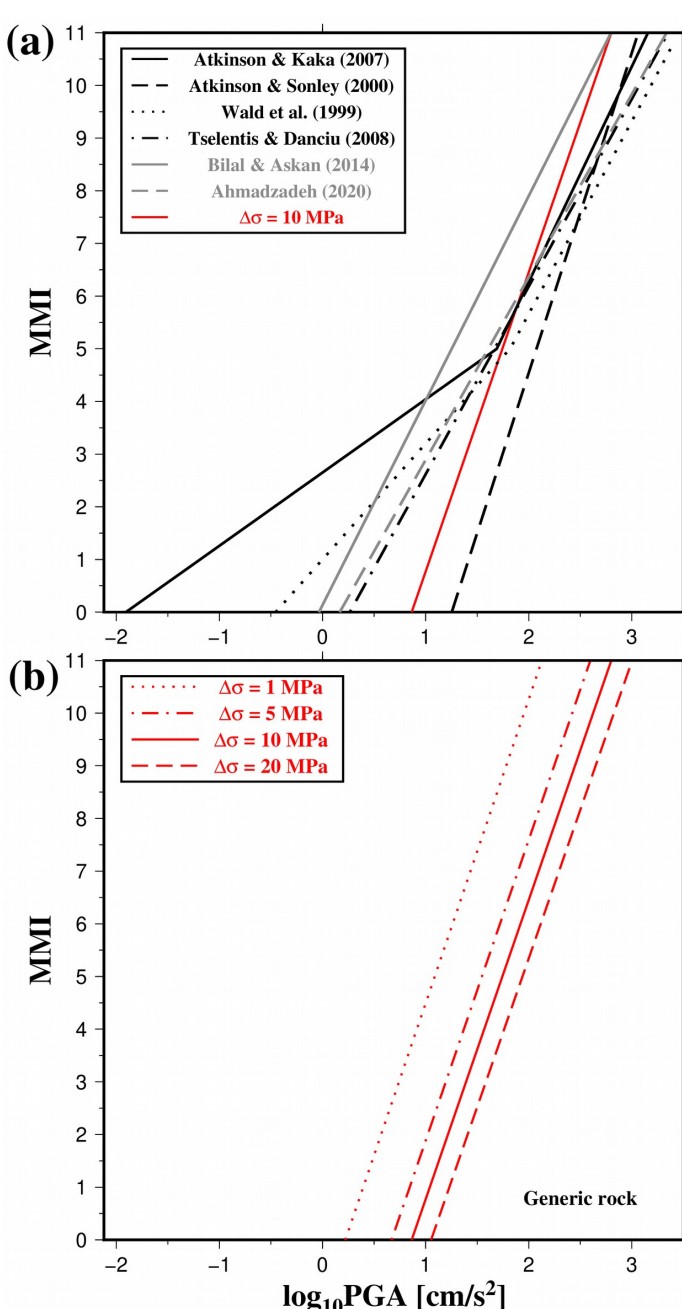

**Figure 9. (a)** Comparison of the obtained MMI-PGA relationships (red lines) considering generic rock

site conditions for the averaged data with previous studies. **(b)** The lower panel shows results for

different stress drop values.


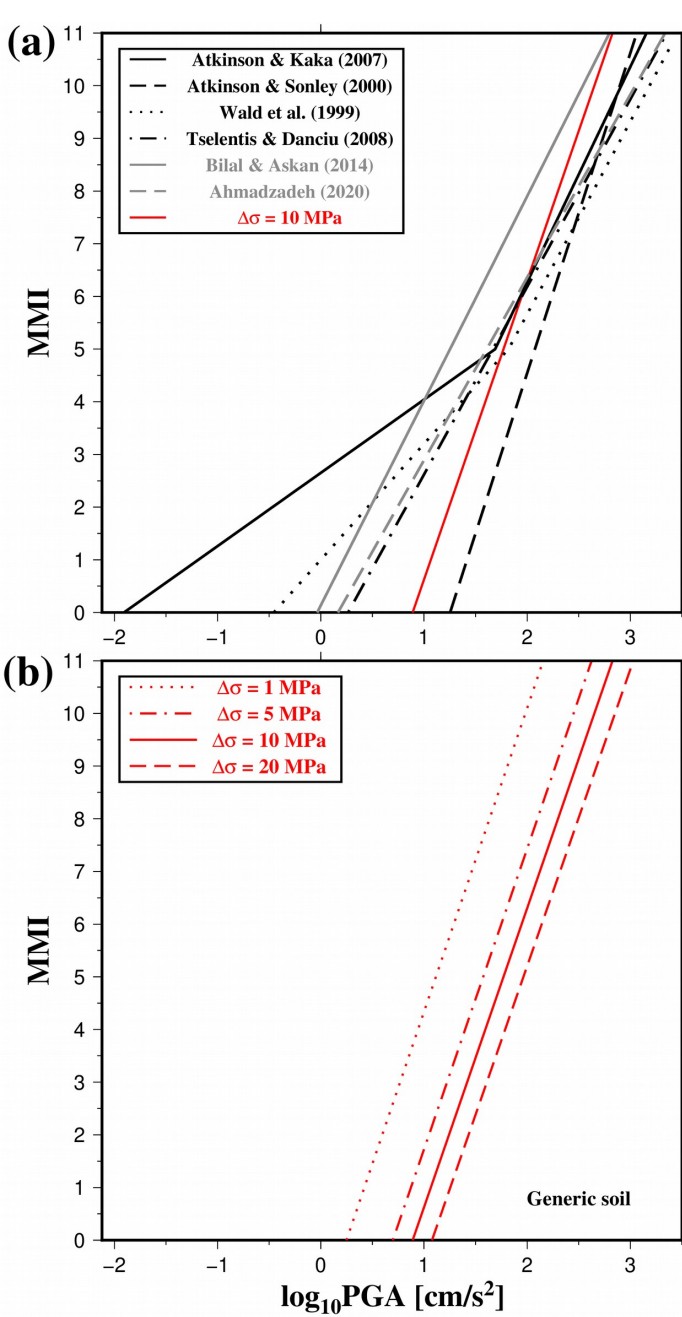

**Figure 10. (a)** Comparison of the obtained MMI-PGA relationships (red lines) considering generic soil

site conditions for the averaged data with previous studies. **(b)** The lower panel shows results for

different stress drop values.





**Table 1.** Studied earthquakes ($M_W$ – moment magnitude)

| Event | Date dd/mm/yyyy | Longitude (°) | Latitude (°) | Depth [km] | $M_W$ | Seismic Intensity Data |
|---|---|---|---|---|---|---|
| 1 | 27/12/1568 | -103.74 | 20.43 | 15 | 7.2 | Suter (2015) |
| 2 | 15/04/1611 | -103.60 | 19.60 | 15 | 6.4 | Suarez (2021) |
| 3 | 23/10/1749 | -103.70 | 20.00 | 15 | 6.4 | Suter (2019) |
| 4 | 12/12/1771 | -103.50 | 20.90 | 10 | 5.5 | Suarez (2021) |
| 5 | 06/11/1774 | -103.73 | 22.04 | 10 | 5.7 | Suter (2020) |
| 6 | 02/10/1847 | -102.84 | 20.36 | 10 | 5.4 | Suter (2018) |
| 7 | 19/06/1858 | -101.00 | 19.50 | 15 | 7.5 | Suarez (2021) |
| 8 | 11/02/1875 | -103.20 | 21.10 | 15 | 6.8 | Suarez (2021) |
| 9 | 05/03/1887 | -109.16 | 30.81 | 18 | 7.5 | DuBois and Smith (1980) |
| 10 | 26/11/1887 | -99.63 | 21.14 | 10 | 5.5 | Suter et al. (1996) |
| 11 | 19/12/1912 | -99.92 | 19.83 | 15 | 6.7 | Suter et al. (1996) |
| 12 | 14/07/1934 | -100.75 | 20.65 | 10 | 4.8 | Suter and Morelos-Rodriguez (2023) |
| 13 | 04/01/1920 | -97.08 | 19.27 | 12 | 6.2 | Suter et al. (1996) |
| 14 | 11/03/1950 | -98.97 | 20.35 | 10 | 5.2 | Suter et al. (1996) |
| 15 | 25/03/1976 | -99.09 | 20.62 | 10 | 5.5 | Suter et al. (1996) |
| 16 | 22/02/1979 | -100.18 | 19.89 | 8 | 5.5 | Suter et al. (1996) |
| 17 | 27/01/1987 | -99.21 | 20.31 | 10 | 4.5 | Suter et al. (1996) |
| 18 | 10/09/1989 | -99.43 | 21.04 | 10 | 4.9 | Suter et al. (1996) |






**Table 2.** Input parameters for finite-fault stochastic ground motion simulations ($Q(f)$ – attenuation
model, κ – spectral decay parameter, β – $S$-wave velocity, ρ – density, $L$ – fault length, $\Delta L$ – subfault
length, $W$ – fault width, $\Delta W$ – subfault width, $\varphi$ – strike, $\theta$ – dip, $\Delta\sigma$ – stress drop)

| N | $Q(f)$ | κ | β [km/s] | ρ [kg/m³] | $L$ [km] | $\Delta L$ [km] | $W$ [km] | $\Delta W$ [km] | $\varphi$ [°] | $\theta$ [°] | $\Delta\sigma$ [MPa] | Site conditions |
|---|--------|---|----------|-----------|----------|-----------------|----------|-----------------|---------------|--------------|----------------------|-----------------|
| 1 | $113f^{0.98}$ | 0.03 | 3.30 | 2360 | 60 | 6 | 30 | 6 | 150 | 50 | 1 5 10 20 | generic rock/soil |
| 2 | $113f^{0.98}$ | 0.03 | 3.58 | 2416 | 24 | 3 | 16 | 4 | 5 | 60 | 1 5 10 20 | generic rock/soil |
| 3 | $113f^{0.98}$ | 0.03 | 3.31 | 2362 | 24 | 3 | 16 | 4 | 10 | 70 | 1 5 10 20 | generic rock/soil |
| 4 | $113f^{0.98}$ | 0.03 | 3.75 | 2450 | 9 | 3 | 9 | 3 | 200 | 50 | 1 5 10 20 | generic rock/soil |
| 5 | $113f^{0.98}$ | 0.03 | 3.59 | 2418 | 10 | 2 | 10 | 2 | 15 | 60 | 1 5 10 20 | generic rock/soil |
| 6 | $113f^{0.98}$ | 0.03 | 3.73 | 2446 | 8 | 2 | 8 | 2 | 87 | 55 | 1 5 10 20 | generic rock/soil |
| 7 | $142f^{1.20}$ | 0.03 | 3.27 | 2354 | 80 | 8 | 40 | 8 | 265 | 70 | 1 5 10 20 | generic rock/soil |
| 8 | $113f^{0.98}$ | 0.03 | 3.48 | 2396 | 40 | 4 | 20 | 4 | 165 | 50 | 1 5 10 20 | generic rock/soil |
| 9 | $84f^{0.90}$ | 0.04 | 3.54 | 2408 | 80 | 8 | 40 | 8 | 12 | 70 | 1 5 10 20 | generic rock/soil |
| 10 | $107f^{0.98}$ | 0.04 | 3.50 | 2400 | 9 | 3 | 9 | 3 | 10 | 50 | 1 5 10 20 | generic rock/soil |
| 11 | $107f^{0.98}$ | 0.04 | 3.45 | 2390 | 30 | 3 | 20 | 4 | 296 | 60 | 1 5 10 20 | generic rock/soil |
| 12 | $107f^{0.98}$ | 0.04 | 3.41 | 2382 | 4 | 1 | 5 | 1 | 5 | 65 | 1 5 10 20 | generic rock/soil |
| 13 | $186f^{0.86}$ | 0.03 | 3.68 | 2436 | 18 | 3 | 15 | 3 | 72 | 50 | 1 5 10 20 | generic rock/soil |
| 14 | $107f^{0.98}$ | 0.04 | 3.47 | 2394 | 6 | 1 | 7 | 1 | 270 | 50 | 1 5 10 20 | generic rock/soil |
| 15 | $107f^{0.98}$ | 0.04 | 3.39 | 2378 | 9 | 3 | 9 | 3 | 270 | 50 | 1 5 10 20 | generic rock/soil |
| 16 | $107f^{0.98}$ | 0.04 | 3.45 | 2390 | 9 | 3 | 9 | 3 | 280 | 66 | 1 5 10 20 | generic rock/soil |
| 17 | $107f^{0.98}$ | 0.04 | 3.58 | 2416 | 3 | 1 | 4 | 1 | 270 | 50 | 1 5 10 20 | generic rock/soil |
| 18 | $107f^{0.98}$ | 0.04 | 3.36 | 2372 | 5 | 1 | 6 | 1 | 10 | 50 | 1 5 10 20 | generic rock/soil |






**Table 3.** Results of regressions (GR – generic rock; GS – generic soil; $\Delta\sigma$ – stress drop; $c_1$, $c_2$, $c_3$, and $c_4$
– regression coefficients; $t_1$ – intersection of the bilinear model; $\sigma_{12}$ and $\sigma_{34}$ – standard errors of
equations)

| Model | Site | $\Delta\sigma$ [MPa] | $c_1$ | $c_2$ | $c_3$ | $c_4$ | $t_1$ | $\sigma_{12}$ | $\sigma_{34}$ |
|---|---|---|---|---|---|---|---|---|---|
| 1 | GR | 1 | 4.27 | 0.34 | 3.33 | 2.40 | 0.46 | 0.98 | 1.61 |
|   |    | 5 | 4.12 | 0.31 | 2.25 | 2.39 | 0.90 | 0.92 | 1.61 |
|   |    | 10 | 4.06 | 0.31 | 1.78 | 2.38 | 1.10 | 0.92 | 1.61 |
|   |    | 20 | 4.00 | 0.32 | 1.31 | 2.38 | 1.30 | 0.86 | 1.61 |
| 1 | GS | 1 | 4.32 | 0.50 | 3.19 | 2.46 | 0.58 | 1.22 | 1.60 |
|   |    | 5 | 4.11 | 0.32 | 2.18 | 2.39 | 0.93 | 0.92 | 1.61 |
|   |    | 10 | 4.05 | 0.32 | 1.72 | 2.38 | 1.13 | 0.91 | 1.62 |
|   |    | 20 | 3.99 | 0.32 | 1.25 | 2.38 | 1.33 | 0.91 | 1.61 |
| 2 | GR | 1 | -1.28 | 5.77 |  |  |  | 0.55 |  |
|   |    | 5 | -3.83 | 5.71 |  |  |  | 0.53 |  |
|   |    | 10 | -4.91 | 5.68 |  |  |  | 0.52 |  |
|   |    | 20 | -5.94 | 5.64 |  |  |  | 0.51 |  |
| 2 | GS | 1 | -1.44 | 5.77 |  |  |  | 0.55 |  |
|   |    | 5 | -3.99 | 5.71 |  |  |  | 0.53 |  |
|   |    | 10 | -5.06 | 5.68 |  |  |  | 0.51 |  |
|   |    | 20 | -6.08 | 5.64 |  |  |  | 0.50 |  |



**Table 4.** Results of regressions for the magnitude and distance residual correction term ($\Delta_{MMI}$) (GR – generic rock; GS – generic soil; $\Delta\sigma$ – stress drop; $c_5$, $c_6$, $c_7$, $c_8$, $c_9$, and $c_{10}$ – regression coefficients; $t_1$ – intersection of the bilinear model; $\sigma_{12}$ and $\sigma_{34}$ – standard errors of equations)

| Model | Site | $\Delta\sigma$ [MPa] | $c_5$ | $c_6$ | $c_7$ | $c_8$ | $c_9$ | $c_{10}$ | $t_1$ | $\sigma_{12}$ | $\sigma_{34}$ |
|---|---|---|---|---|---|---|---|---|---|---|---|
| 1 | GR | 1 | 2.46 | -0.54 | 0.15 | -0.17 | 0.06 | -0.10 | 0.46 | 0.96 | 1.61 |
| | | 5 | 1.87 | -0.40 | 0.07 | -0.17 | 0.05 | -0.09 | 0.90 | 0.91 | 1.61 |
| | | 10 | 1.88 | -0.40 | 0.07 | -0.17 | 0.06 | -0.09 | 1.10 | 0.91 | 1.61 |
| | | 20 | 2.25 | -0.43 | -0.08 | -0.23 | 0.06 | -0.08 | 1.30 | 0.84 | 1.61 |
| 1 | GS | 1 | 3.31 | -0.64 | -0.06 | -0.29 | 0.08 | -0.10 | 0.58 | 1.19 | 1.60 |
| | | 5 | 1.89 | -0.40 | 0.06 | -0.17 | 0.05 | -0.09 | 0.93 | 0.91 | 1.61 |
| | | 10 | 1.95 | -0.41 | 0.07 | -0.18 | 0.06 | -0.09 | 1.13 | 0.90 | 1.61 |
| | | 20 | 1.96 | -0.41 | 0.06 | -0.19 | 0.06 | -0.09 | 1.33 | 0.90 | 1.61 |
| 2 | GR | 1 | 1.62 | -0.47 | 0.70 | | | | | 0.53 | |
| | | 5 | 1.46 | -0.45 | 0.75 | | | | | 0.51 | |
| | | 10 | 1.40 | -0.45 | 0.77 | | | | | 0.50 | |
| | | 20 | 1.29 | -0.44 | 0.80 | | | | | 0.49 | |
| 2 | GS | 1 | 1.59 | -0.47 | 0.74 | | | | | 0.53 | |
| | | 5 | 1.42 | -0.46 | 0.79 | | | | | 0.51 | |
| | | 10 | 1.35 | -0.45 | 0.81 | | | | | 0.50 | |
| | | 20 | 1.25 | -0.44 | 0.85 | | | | | 0.49 | |