# Peer review of "Conversion relationships between Modified Mercalli Intensity and Peak Ground Acceleration for historical shallow crustal earthquakes in Mexico 3"

_Natural Hazards and Earth System Sciences, 2024_

## Author Comment (AC3)

**Response to comments by editor and reviewers.**

We appreciate the comments from the reviewers, which have allowed us to improve our manuscript. Overall, we have followed all the suggestions. In the following we provide a response to the individual points raised. Changes made to the manuscript are highlighted in yellow.

**Editor.**

**The manuscript entitled 'Conversion relationships between Modified Mercalli Intensity and Peak Ground Acceleration for historical shallow crustal earthquakes in Mexico' proposed new relationships between modified Mercalli intensity and peak ground acceleration (PGA) for shallow crustal earthquakes in Mexico. Such a study could be crucial for subsequent research on seismic hazards and for determining strategies for hazard mitigation. After read through this manuscript, I have a number of questions and comments about the manuscript, listed below.**

**Source parameter uncertainty: Earthquake simulation requires several parameters. Although the rupture parameters for each earthquake was shown in Tables 1 and 2, their reliability and uncertainty were not discussed. I also noticed that not all of parameters shown in equation (2) were specified in the text (e.g., S-wave velocity, corner frequency). Furthermore, it is rather difficult to obtain rupture parameters for historical events (without instrumental records).**

S-wave velocity is reported in Table 2 with the Greek letter $\beta$. Corner frequencies ($f_c$) were estimated with the following relationship: $\log_{10} (f_c) = 1.32 + 0.33 \log_{10} (\Delta\sigma) + \log_{10} (\beta) - 0.5 M_W$ (Havskov and Ottemöller, 2010) which is now specially mentioned.

**Simulation reliability: The uncertainties of the source parameters were not appropriately addressed, and the reliability of the simulations was also not thoroughly discussed. While considering different stress drops is a good approach, the study only presented the results with the lowest deviation for the regression with intensity. In my opinion, this comparison alone cannot accurately identify the proper stress drop. Instead, I suggest obtaining the stress drop for some events from previous studies or waveform inversion. Additionally, there is no discussion on confirming the simulation's reliability. I recommend comparing the simulated PGA with some instrumental observations to validate the simulations.**

We simulated three events that were instrumentally recorded to validate our results. In that process, we determined spectral source parameters, especially the stress drop. An example is shown in the supplementary materials. We have previously used

the stochastic method to simulate ground motions of Mexican earthquakes with consistent results that agree with observations, confirming that with the correct selection of input parameters, the method is able to reproduce ground motion characteristics. For example:

-Rodríguez-Pérez Quetzalcoatl, Lars Ottemöller, Raúl R. Castro; Stochastic Finite-Fault Ground-Motion Simulation and Source Characterization of the 4 April 2010 Mw 7.2 El Mayor-Cucapah Earthquake. Seismological Research Letters 2012;; 83 (2): 235–249. doi: https://doi.org/10.1785/gssrl.83.2.235.

-Rodríguez-Pérez, Q., Ottemöller, L. & Raeesi, M. Source study of three moderate size recent earthquakes in the Guerrero seismic gap. J Seismol 19, 753–780 (2015). https://doi.org/10.1007/s10950-015-9493-0

**Path and site effect: Equation (1) represents the source, site, and path components in the waveform simulation. While the source component has been described in the text, the path and site effects have not been specified. Which velocity model was implemented, and are the simulation results sensitive to the velocity model? Although this study proposed simulation sets incorporating both rock and soil conditions, the epistemic uncertainty could be further minimized by considering the specific site conditions for each location.**

The method used the S-wave velocity at the hypocenter, and it is incorporated in the source term. These values were taken from an average elastic model of Spica et al. (2016). Variations in S-wave velocity is not so drastic, as all the events are shallow. In the case of a deep earthquake variations will be more important. On the other hand, we do not have information about site effects in the locations where MMI was reported. For this reason, we adopted representative models proposed by Boore and Joyner (1997) to provide intervals of PGA estimated by considering rock and soil conditions.

**Reviewer 1.**

**This manuscript is well written and easy to read. The new empirical relations proposed will be useful for future studies. The only weakness that I find in this study is the lack of actual PGA observations that could be useful to calibrate the stochastic models. However, the parameters used with EXSIM12 are adequate and provide conservative estimates of PGA.**

**A few suggestions follow:**

**Pg.3, line 24: the 1887 event is better known as the Sonora earthquake.**

We corrected this sentence.

**Pg.6, line 18: I think the year of the reference is 2015.**

According to the authors, to cite the code, the year is 2012.

**Pg.7, lines 10-11: Are the stress drop values (1, 5, 10 and 20MPa) typical or expected for the earthquakes modeled?**

Yes, these stress drop values were considered as representative for shallow crustal intraplate events based on previous studies.

**Pg.7, line 17: is there any observed PGA information from any of the earthquakes analyzed or from other similar events?**

Unfortunately, there are few records for shallow crustal events in the Trans-Mexican Volcanic Belt and in North of Mexico. Many of the records are classified as low quality signals for the seismological agencies.

**Pg.9, lines 19-20: "The bilinear models differ significantly at lower PGA values" Why is that?**

These discrepancies in low PGA values are related to inaccurate input parameters to describe path effects at long distances.

**Reviewer 2**

**Rodríguez-Pérez and Zúñiga have proposed PGA to MMI relationships for Mexico, by using observed MMI from historical events, and synthetic PGA simulated using stochastic finite-fault method. The author's PGA simulation approach appears comprehensive. This study after some improvements, could be a good contribution to the seismic research community.**

**However, I have some major concerns in the study in its current form, especially relating to the statistical analysis. One of the primary challenges for this study is not being able to fully validate its results due to lack of ground truth PGA observations. I have described all my concerns below. As line numbers restart on each page, I have used the notation pagenum.linenum.**

**1.General - The problem that the authors are trying to solve has not been defined anywhere, including the abstract and introduction. They mention that the goal of the study is to develop new empirical relationships between MMI and PGA in Mexico. But why is it necessary to develop these relationships? It would be helpful to include text to motivate the problem that the authors are trying to solve.**

> Yes, we agree with the reviewer. We have included the fact that it is necessary to develop conversion relations between MMI and PGA, to determine intensity maps for previous and future scenarios. We also modified the abstract and introduction.

**2.General - One sentence that could be the potential motivation for this study is Line 10.13 where the authors mention that reported MMI could be used as a proxy for estimating PGA for historical earthquakes using their developed relationships, in order to better understand seismic mechanisms in Mexico, and eventually estimate better hazard. However, throughout the regression analysis, the authors have used MMI as the dependent variable and PGA as the independent variable, which would not meet this objective of developing a relationship to convert MMI to PGA. Instead, the regression analysis should be done with PGA as the dependent variable.**

Regarding the form of the regression equations, the vast majority of the conversion relationships between MMI and PGA have the form: MMI = f(PGA) where MMI is the dependent variable, while marginally in the form PGA = f(MMI) where PGA is dependent the variable, for this reason, we adopted the first form, in this way, we can ffectively compare our results with studies with similar tectonic conditions. Nevertheless, we have added relations in its inverse form.

**3.Line 1.21 - I could not find support in the text for this statement in the abstract - The refined PGA to MMI conversion equations show slightly less variability than simple linear equations in predicting intensity values. Additionally, it is not clear what the authors are referring to as "refined" and "simple" equations in the statement.**

> We rewrote this section to clarify our  meaning.

**4.Line 1.25 - The statement could be made better by adding how "the proposed relationships can be used for improved hazard assessments in Mexico".**

> We modified this sentence.

**5.Line 3.3 - Suggest changing sentence for clarity - Various studies have proposed relationships between seismic intensity (measured by MMI, DYFI, JMA, etc.) and ground motion parameters (measured by PGA, PGV, Sa, etc.) across distinct geographical regions.**

> We modified this sentence.

**6.Line 3.11 - Since the last M ≥ 6 event \*that\* originated in north...**

    We corrected this sentence.

**7.Fig 1 - The dark gray shading, dashed lines, and line with triangles on the map are undefined.**

    We provided a description of the information shown in figure 1.

**8.Fig 1 - The figure can be improved by changing the color/style of earthquakes in each of the three regions, in order to easily associate events with their respective regions.**

We now use different colors for each of the tectonic regions.

**9.Fig 1 - The figure can be further improved by using a topographic map as base layer.**

    We decided to keep the figure in its original form so that it does not appear to be clogged up.

**10.Paragraph 4.19 - The information currently provided for observed MMI reports is insufficient. Please provide more information about how MMI was calculated for historical earthquakes, especially those dating back to the 1500's, and the sources of original data for these MMI reports.**

    The seismic intensity estimates are supported by a large number of historical and bibliographic sources in Mexico and abroad (specific details of the historical sources can be found in Suter et al., 1996; Suter, 2015, 2018, 2019, 2020, 2021; Suter and Morelos-Rodriguez, 2023; listed in the references). In the case of the MMI data taken from the catalog of Suarez (2021), its seismic intensity estimations are based on the historical research of Mexican earthquakes compiled by García-Acosta and Suarez (1999). The information about the MMI reports is given in section 3 data and methods.

**11.Paragraph 4.19 - It would be helpful to include a figure with the count and range of MMI reports obtained for each of the 18 events used in this study. Although this could be roughly interpreted by combining Fig 2 and Table 1, adding a separate figure will both improve the data section, and make it easier for the reader to understand the observation data.**

    We now present this information in table 1.

**12.Line 4.20 - Suggest specifying "moment magnitudes".**

    We clarified this information in the text.

**13.Eq 2 - Suggest including the left side of the equation. Y = ...**

    We did not modify this equation. In its actual form, the reader can easily identify the variables of equation 2.

**14.Eq 2 - kappa is undefined.**

    We now define kappa in the text.

**15.Eq 2 - G(r) is undefined. Perhaps, it's supposed to be G(R).**

Yes, the reviewer is right. We have corrected equation 2.

**16. Line 5.23 - R_{theta phi} is not present in the equation. theta and phi are undefined.**

We corrected equation 5. $R_{\theta\phi}$ is defined as the radiation pattern.

**17. Line 6.4 - Suggest defining i, j prior to this line for better clarity.**

i and j denote the subfault along the length and width of the rupture plane respectively. We have added this text.

**18. Eq 3 - Please define the difference between lowercase and uppercase `a`, and the relationship between them, if any. Lowercase a(t) is defined as the sum of sub fault contributions and it is not clear which contribution it refers to.**

The ground motions of subfaults ($a_{ij}(t)$) are summed with a proper time delay in the time domain to obtain the ground motion acceleration, a(t), from the entire fault. We have added this text.

**19. Line 6.15 - Is the time delay relative to the sub fault i=0, j=0?**

The rupture is assumed to nucleate from the hypocenter and propagate with a constant rupture velocity. The rupture front reaches each sub-fault with a delay that depends on its location relative to the hypocenter and on rupture velocity.

**20. Line 6.15 - I am confused about the notation i, j. In eq 3, it is used as sub fault along the length and width of the rupture plane respectively, but has not been defined as such in previous equations. Are these i, j different than the ones in eq 2?**

No, in both equations i and j are used to define subfaults of the rupture plane.

**21. Line 6.16 - I am not familiar with the term "rise time". Could this be defined?**

We defined the rise time in the text.

**22. Line 7.4 - It will be helpful to mark the 1887 event on the map in Fig 1 for better understanding of its location with respect to other events.**

We modified figure 1 accordingly.

**23. Line 7.11 - From Figs 3 and 4, it appears that the PGA for different stress drops was calculated separately and regression equations were developed for each stress drop. It would be helpful to clarify that here, and also add why multiple values were selected, compared to all other variables, for which the best single value was chosen.**

We clarified the selection of stress drop as pointed out by both reviewers.

**24. Line 7.22 - It is not clear where two branched trend was "previously reported".**

We clarified this point in the text.

**25. Line 7.23 - The difference between Figs 3 and 4 is never described.**

We now mention the difference between Figs 3 and 4 in the manuscript.

**26.Line 8.2 - MMI defined up to two decimal points indicates high precision in MMI reporting, which seems unlikely given the historical nature of most events. My previous comment about providing more information about MMI calculation might help here. Overall, it would be helpful to understand the expected variability in the MMI estimates from the reports.**

Unfortunately, MMI estimations for Mexico do not report uncertainties. We have answered the other comment.

**27.Paragraph 7.17 - The statements regarding the two methods - using complete dataset and MMI bucket-averaged datasets are mixed together in this paragraph, making it harder to understand which model the authors are referring to in a given sentence. Suggest splitting the two models in different paragraphs for clarity.**

We rewrote this section.

**28.Eqs 4 and 5 - The number `1` in t_1 indicates presence of t_x. Suggest removing the numeral since there is only one threshold in the two branch model.**

We removed subindex 1 as suggested by the reviewer.

**29.Eq 5 - logPGA is not in the equations but is used in the threshold t_1. Instead, since the correction term is a function of MMI, the threshold should be changed to MMI in this equation. For example, delta_MMI (MMI) = ... for MMI < x.**

The threshold is the same of equation 4 that is the reason why we keep this value.

**30.Eq 5 - If the MMI is near threshold t_1, is it possible that applying the correction would move the MMI to a different branch? It would be helpful to clarify this and how to resolve this situation.**

The mathematical form of equations 4 indicate that if PGA value is higher or lower than t a specific equation is applied.

**31.Line 8.20 - The statement is unclear, and I am not sure how the models were combined.**

According to Atkinson and Kaka (2007) the improved relations are defined as

equation (1) + ΔMMI

where Eq 1 is

MMI  C1 + C2 log Y,   log Y < value
MMI  C3 + C4 log Y,   log Y > value

ΔMMI= C4 + C5 M + C6 log R,

Resulting in

MMI  C1 + C2 log Y + C5 + C6 M + C7 log R,   log Y < value

MMI  C3 + C4 log Y + C5 + C5 M + C7 log R,   log Y > value

**32.Tables 3, 4 - Please define standard errors in equations 4 and 5 as it is difficult to associate them to the respective equations in their current form. For example, MMI=c1+c2 logPGA + epsilon, where epsilon ~ N(0, sigma_ij)**

   We clarified this point in the manuscript.

**33.Line 9.3 - Please describe how the "evaluation" was done. There appears to be lower residual variability at lower magnitudes, and both low and high hypocentral distances from Fig 5.**

   We compared the magnitud of the residuals for linear and bilinear models. We agree with the reviewer concerning the differences between the two datasets. The residual variability at lower magnitudes is no clear due to large dispersion in the data. We clarified this point in the text.

**34.Line 9.5 - The sentence is not clear. What is meant by residuals are smaller by factor of 8 - is it the variability in residuals, or the extreme residual value from bilinear model compared with that of the averaged model?**

   We now present the results for each case as they came from different datasets.

**35.Line 9.5 - Since there is only one data point per MMI (due to averaging PGA) for model 2, the comparison between models 1 and 2 cannot be made directly. The variability of residuals given M, R will always be zero or close to zero for model 2 due to the very few data points. In fact, it is more likely that the mean of residuals is non-zero for model 2 given the scarcity of data points, thus nullifying the assumption of residuals following zero-mean normal distribution.**

We agree with the reviewer that comparison cannot be made directly, so we present results separately,

**36.Figs 7-10 - It would be better to combine subfigures a and b to be able to compare the results of this study with existing relationships across all proposed stress drops. In fact, it may be helpful if the subfigures were shown for bilinear and linear models instead, so they can be compared easily. Another option will be to show 1 figure per stress drop with both models in the same figure.**

   We did not change the figures because we consider that the information is clear enough.

**37.Line 9.16 - Why were these specific models chosen? Please provide information about comparative model selection based on similar magnitudes/distances/rupture mechanisms/site conditions, etc. In**

**addition, please add a statement about which of the comparative relationships is expected to be the closest to that of this study based on rupture and ground motion attenuation characteristics in Mexico, to make it easier for the reader to focus on the differences with that model.**

We now provided information on this issue in the text.

**38.Line 9.20 - It appears that the difference of bilinear models occurs at t_1 which happens to be close to 10 cm/s2. Suggest modifying this statement to indicate that the lower branch of the bilinear model (approx < 10cm/s2) deviates from other existing relations.**

We modified this statement.

**39.Line 10.1 - I disagree with this and the following statements, as the standard error between models 1 and 2 cannot be compared directly due to the large difference in the number of points used for regression in both models.**

We corrected this statement.

**40.Line 10.9 - This statement may need to be revised based on my previous comment about observing some trends in residuals over magnitudes and distances from the figures.**

In our perspective, the trend among the residuals and magnitudes and distances is not evident.

**41.Line 10.13 - The approach taken by this study is interesting given the authors' statement that the objective is to convert reported MMI to PGA/PGV, instead of converting PGA to MMI. In that sense, why have the authors chosen to use MMI as the dependent variable in this study? Note that linear regression does not give the same result by reversing the dependent variable since the residual square error is minimized for the dependent variable only. In that sense, in order to create an appropriate relationship to convert reported MMI to PGA, the authors should revisit their approach and fit regression equations with PGA as the dependent variable.**

We now incorporated regressions considering PGA as the dependent variable.

**42.Line 10.18 - Please include hypocentral distance for the MMI observation.**

The hypocentral distance is about 25 km.

**43.Line 10.21 - I disagree with the statement that "stochastic ground motion simulation method is a valid methodology", since it indicates to the reader that this may be one of the most appropriate approaches to solve this problem. In fact it is not possible to evaluate the validity of this simulation approach with other simulation approaches since there is no ground truth data. In that sense, it is completely possible**

**to use an entirely different approach (e.g., a simplified GMPE with point sources) to generate synthetic PGA, and then fit regression equations between those PGA and observed MMI. Without ground truth, there would be no way of knowing which of the two approaches are better suited. No doubt that this study is comprehensive and beneficial for pushing the boundaries of seismic research in Mexico, however, the authors have only proposed one of the possible valid methodologies, the validity of which cannot be determined given the study's lack of ground truth MMI + PGA observations at the same sites.**

We are aware of the limitations of the approach used, but it can represent an option for studying historical earthquakes in conditions of lack of instrumental data, as is our case.